# Do recent advancements in model-based deep reinforcement learning really improve data efficiency?

## Abstract

Reinforcement learning (RL) has seen great advancements in the past few years. Nevertheless, the consensus among the RL community is that currently used model-free methods, despite all their benefits, suffer from extreme data inefficiency. To circumvent this problem, novel model-based approaches were introduced that often claim to be much more efficient than their model-free counterparts. In this paper, however, we demonstrate that the state-of-the-art model-free Rainbow DQN algorithm can be trained using a much smaller number of samples than it is commonly reported. By simply allowing the algorithm to execute network updates more frequently we manage to reach similar or better results than existing model-based techniques, at a fraction of complexity and computational costs. Furthermore, based on the outcomes of the study, we argue that the agent similar to the modified Rainbow DQN that is presented in this paper should be used as a baseline for any future work aimed at improving sample efficiency of deep reinforcement learning.

## 1 Introduction

Producing fully independent agents that learn optimal behavior and develop over time purely by trial and error interaction with the surrounding environment is one of the prominent dilemmas in the field of artificial intelligence. A mathematical framework that encapsulates the problem of these autonomous systems is reinforcement learning. Over the past few years, exceptional progress has been made in devising artificial agents that can learn and solve problems in a variety of domains using deep RL methods (Mnih et al., 2015; Schulman et al., 2015; Silver et al., 2016). However, these algorithms are perceived as extremely data inefficient. They are thought to require an immense amount of non-optimal interaction with the real environment before they begin to operate acceptably well (Irpan, 2018).

One of the most popular benchmarks for assessing overall performance and data complexity of deep RL algorithms is Atari Learning Environment (Bellemare et al., 2013; Machado et al., 2018). The state-of-the-art model-free approaches, at least in the way they were presented so far, need millions of frames to learn how to play these games acceptably well (Schulman et al., 2017; Hessel et al., 2018). It corresponds to days of play experience using the standard frame rate. However, human players can achieve the same within minutes (Tsividis et al., 2017).

A lot of work has been produced to circumvent these shortcomings. Most successful studies focus on model-based strategies inspired by the classical Dyna approach (Sutton, 1991) and action-conditional prediction methods (Oh et al., 2015; Leibfried et al., 2016). Although some of them manage to drastically reduce the amount of data required by the standard algorithms, they do it by highly increasing both conceptual and computational complexity of the models.

In this paper, we argue, and experimentally prove, that already existing model-free techniques can be much more data-efficient than it is assumed. We introduce simple change to the state-of-the-art Rainbow DQN algorithm. In some environments like Pong or Hero, it can achieve the same results given only 5% - 10% of the data it is often presented to need. Furthermore, it results in the same data-efficiency as the state-of-the-art model-based approaches while being much more stable, simpler, and requiring much less computation.

Following the introduction, section 2 gives a brief background behind reinforcement learning with the focus on Q-learning and its deep learning equivalents. Section 3 provides an overview of recent studies aimed at improving data efficiency using model-based approaches. Section 4 argues that model-free methods can be much more efficient than it tends to be presented and that existing model-based techniques only give an illusion of efficiency. Then, the description and analysis of experiments follow in sections 5 and 6. Finally, section 7 concludes this study.

## 2 BACKGROUND

Reinforcement learning is a problem of learning a policy that maximises the reward signal for a given task. To define RL setting we need a set of possible environment states $\mathbb{S}$, a set of available actions $\mathbb{A}$, and relations between those. These relations are described by a transition function $T : \mathbb{S} \times \mathbb{A} \to \mathbb{S}$ that defines dynamics of transitions from one state to another, and a reward function $R : \mathbb{S} \times \mathbb{A} \to \mathbb{R}$ that defines the real-valued reward signal. Together, $T$ and $R$ constitute the model of the environment. The goal of reinforcement learning is to find a policy $\pi : \mathbb{S} \to \mathbb{A}$ that maximises the total cumulative reward over time. One of the most popular reinforcement learning algorithms is Q-learning (Watkins & Dayan, 1992). Q-learning decides on an optimal policy based on the state-action value function $Q : \mathbb{S} \times \mathbb{A} \to \mathbb{R}$ that maps state and action performed in that state to the expected total cumulative reward following the action. The algorithm chooses an action that maximizes $Q$, i.e. $a_t = \text{argmax}_a Q(s_t, a)$. $Q$ is learned in the process of interacting with the environment. At every agent's step, tuple $(s_t \in \mathbb{S}, a_t \in \mathbb{A}, r_t \in \mathbb{R}, s_{t+1} \in \mathbb{S})$ is obtained and immediately used to update the Q function. Because state-action combination is often too big or continuous to represent directly in a tabular manner, $Q$ is commonly approximated using different supervised learning algorithms. However, using deep learning to approximate $Q$ is not trivial because Q-learning breaks important assumptions required by neural networks. Namely, $Q$ update is recursive and experience tuples are highly correlated when used sequentially.

Recently introduced DQN (Mnih et al., 2015) bypassed this issue by introducing two concepts: target network and replay buffer. Target network is simply a fixed snapshot of the network that approximates $Q$ value (online network) taken every $\tau_t$ steps. Instead of updating the online network towards itself, it is updated towards the target network. This approach maintains the logic of Q-learning while stopping the online network from diverging due to recursive updates. Replay buffer, on the other hand, guarantees a much higher level of independence between experience tuples. They are not used immediately, one after another anymore but stored in the replay buffer instead. Then, every $\tau_u$ steps, a single training step is performed, i.e. a mini-batch of randomly sampled experience from the replay buffer is used to update the online network. It reduces the correlation between experience samples by breaking their ordering.

Rainbow DQN (Hessel et al., 2018) is a combination of several incremental improvements on top of DQN that increased both sample efficiency and the total performance of the algorithm achieving state-of-the-art results. It is an architecture that we use as an example that current model-free deep RL is not as inefficient as it is often stated. Throughout the paper hyperparameters from Hessel et al. (2018) are employed, unless stated otherwise.

## 3 MODEL-BASED REINFORCEMENT LEARNING

The most successful approach to improving data efficiency of deep RL is based on the premise of model-based techniques (Sutton & Barto, 2018). Having access to transition and reward mechanics of the environment would make it possible to construct an artificial simulation where the agent could be trained without performing often costly interactions with the real environment. However, in most scenarios, the agent is not given any prior information about the model of its environment. This issue is often overcome by learning the model instead. Oh et al. (2015) and Leibfried et al. (2016) have shown that it is possible with a very high level of accuracy.

Ability to learn the model of the environment was subsequently leveraged to successfully improve different aspects of deep RL (Racanière et al., 2017; Oh et al., 2017; Buesing et al., 2018; Ha & Schmidhuber, 2018). Azizzadenesheli et al. (2018), Holland et al. (2018), and Kaiser et al. (2019), however, focused directly on employing the learned models to increase data efficiency of deep RL algorithms.

Azizzadenesheli et al. (2018) proposed Generative Adversarial Tree Search (GATS). Unlike in the standard approach to learning the environment dynamics, GATS creates two separate models: Generative Dynamics Model (GDM) based on a modified Pix2Pix (Isola et al., 2017) to learn the transition model $T : \mathbb{S} \times \mathbb{A} \rightarrow \mathbb{S}$; and Reward Predictor (RP), a simple 3-class classification architecture to learn the reward model $R : \mathbb{S} \times \mathbb{A} \rightarrow \mathbb{R}$. Both models learn from experience stored in DQN's replay buffer and are then used for bounded Monte Carlo tree search as in (Silver et al., 2016). GATS is evaluated primarily on the game Pong where it learns an optimal policy using around 42% of the data required by using standalone model-free agent what is a tiny improvement compared to the methods described next.

Holland et al. (2018) explored the performance of the model-based approach given either perfect model, model pretrained on expert data (pretrained model), or model learned alongside the agent's value function (online model). Both non-perfect models followed standard architecture for the task (Oh et al., 2015; Leibfried et al., 2016). These models are then used to generate 100 samples of simulated experience for every interaction with the real environment. All three variations outperformed state-of-the-art Rainbow DQN in terms of data efficiency on 5 out of 6 games. Nevertheless, only the results of the online model are used for further discussion to ensure a fair comparison between the algorithms.

Kaiser et al. (2019) introduced Simulated Policy Learning (SimPLe). Similarly to the previous two architectures, it learns the model of the environment using a modified version of Oh et al. (2015). It differs from previous approaches by employing PPO (Schulman et al., 2017) as its RL agent and by using the learned model much more exhaustively. It uses the model similarly to Holland et al. (2018), however it provides at least 800k samples of artificial data after every 6.4k interactions. The approach is then evaluated on a range of 26 different Atari games. It provides results that highly outperform both Holland et al. (2018) and Azizzadenesheli et al. (2018) in terms of data efficiency achieving at least 2x improvement on over half of the games and more than 10x improvement on Freeway. To the best of our knowledge, SimPLe is the state of the art in terms of data-efficient deep reinforcement learning; thus, it will be used as a primary baseline throughout the rest of the paper.

## 4 DATA EFFICIENCY OF STANDARD APPROACHES

We argue that DQN-like model-free methods are not as data inefficient as they are often portrayed. They are simply used in a very inefficient way. Let us define ratio $r$ describing the number of training steps to the number of interactions with the environment. In the default setting $\tau_u = 4$. It means that the algorithm performs a single update of the network for every 4 interactions with the environment, i.e., $r = 1/4$.

As explained in section 3, both the online-model-based algorithm from Holland et al. (2018) and SimPLe from Kaiser et al. (2019) first learn the approximated model of the environment. Then, this approximation is used to provide simulated samples of experience alongside the real data. Nevertheless, these samples, in the best case, can only provide as much real signal to the agent as was provided in the original data. However, as a byproduct of the agent's interactions with the learned model, the ratio $r$ significantly increases. Holland et al. (2018) performs 100 simulated steps for each real step causing $r = (1 + 100) * (1/4) = 25.25$. SimPLe executes 800k simulated steps after every 6.4k interactions with the real environment. Thus, if SimPLe was using DQN as its model-free component ratio $r$ would be even higher ($r = (800k + 6.4k)/6.4k/4 = 126/4 = 31.5$).

It seems unfair to allow model-based methods to perform more training steps for each gathered data point without letting model-free baselines to do the same. However, from the studies discussed above, only Holland et al. (2018) performed tests allowing DQN for extra updates[1]. GATS was compared solely to the standard version of DQN and SimPLe to the standard version of PPO algorithm together with the Rainbow DQN that, as stated in the paper, was hypertuned for sample efficiency (HRainbow). However, hyperparameters for HRainbow were not disclosed. We hypothesize, that the main reason behind improved data efficiency in the results is essentially increased $r$.

---

[1]Their results showed that indeed model-based approach with the online model does not overperform model-free approach with extra updates. However, the study was mainly interested in thorough analysis, rather than improving the state of the art.

Table 1: Mean scores produced by each approach in the low-data regime. Scores for OTRainbow, SimPLe, HRainbow, and Rainbow are obtained after 100k interactions with the real environment. Values in bold represent the top model for the game (ignores Human).

| Game | OTRainbow | SimPLe | HRainbow | SRainbow | Human | Random |
|---|---|---|---|---|---|---|
| Alien | **824.7** | 616.9 | 290.6 | 318.7 | 6875 | 184.8 |
| Amidar | **82.8** | 74.3 | 20.8 | 32.5 | 1676 | 11.8 |
| Assault | 351.9 | **527.2** | 285.7 | 231 | 1496 | 248.8 |
| Asterix | 628.5 | **1128.3** | 300.3 | 243.6 | 8503 | 233.7 |
| BankHeist | **182.1** | 34.2 | 34.5 | 15.55 | 734.4 | 15 |
| BattleZone | **4060.6** | 4031.2 | 3363.5 | 3285.71 | 37800 | 2895 |
| Boxing | 2.5 | **7.8** | 0.9 | -24.8 | 4.3 | 0.3 |
| Breakout | 9.84 | **16.4** | 3.3 | 1.2 | 31.8 | 0.9 |
| ChopperCommand | **1033.33** | 979.4 | 776.6 | 120 | 9882 | 671 |
| CrazyClimber | 21327.8 | **62583.6** | 12558.3 | 2254.5 | 35411 | 7339 |
| DemonAttack | **711.8** | 208.1 | 431.6 | 163.6 | 3401 | 140 |
| Freeway | **25** | 16.7 | 0.1 | 0 | 29.6 | 0 |
| Frostbite | 231.6 | **236.9** | 140.1 | 60.2 | 4335 | 74 |
| Gopher | **778** | 596.8 | 748.3 | 431.2 | 2321 | 245.9 |
| Hero | **6458.8** | 2656.6 | 2676.3 | 487 | 25763 | 224.6 |
| Jamesbond | **112.3** | 100.5 | 61.7 | 47.4 | 406.7 | 29.2 |
| Kangaroo | **605.4** | 51.2 | 38.7 | 0 | 3035 | 42 |
| Krull | **3277.9** | 2204.8 | 2978.8 | 1468 | 2395 | 1543.3 |
| KungFuMaster | 5722.2 | **14862.5** | 1019.4 | 0 | 22736 | 616.5 |
| MsPacman | 941.9 | **1480** | 364.3 | 67 | 15693 | 235.2 |
| Pong | 1.3 | **12.8** | -19.5 | -20.6 | 9.3 | -20.4 |
| PrivateEye | **100** | 35 | 42.1 | 0 | 69571 | 26.6 |
| Qbert | 509.3 | **1288.8** | 235.6 | 123.46 | 13455 | 166.1 |
| RoadRunner | 2696.7 | **5640.6** | 524.1 | 1588.46 | 7845 | 0 |
| Seaquest | 286.92 | **683.3** | 206.3 | 131.69 | 20182 | 61.1 |
| UpNDown | 2847.6 | **3350.3** | 1346.3 | 504.6 | 9082 | 488.4 |

## 5 EXPERIMENTAL SETUP

To test the above-mentioned hypothesis, we train a standard Rainbow DQN agent, as described in Hessel et al. (2018), with only a few small differences to increase ratio $r$. Firstly, we decrease period between updates as much as possible so $\tau_u = 1$ (thus $r = 1$). Then, because it is impossible to further increase $r$ using existing hyperparameters, we introduce a new parameter $k$ that specifies how many network updates should be performed every $\tau_u$ steps (similarly to DQN Extra Updates from Holland et al. (2018)). We find that $k = 8$ produces the best results (hence $r = 8$). We also decrease epsilon decay period to only 50K steps to make it compatible with low data settings. We will refer to this modified version of Rainbow DQN as 'OTRainbow' (Overtrained Rainbow) throughout the rest of the paper.

Existing code for the Rainbow DQN from the Dopamine framework (Castro et al., 2018) was modified as explained above to obtain OTRainbow. Dopamine was used for two reasons: (*i*) it allows for quick and easy prototyping of new RL algorithms; (*ii*) to ensure the same implementation for each version of the Rainbow DQN (whether it is OTRainbow, HRainbow, or standard Rainbow). It was then evaluated on the same range of 26 Atari games from the Atari Learning Environment as used by SimPLe in the original paper. We then compare the outcomes to multiple different baselines: an agent that always chooses action uniformly at random (Random), human score as reported by Mnih et al. (2015) (Human), Rainbow DQN with the original hyperparameters from Hessel et al. (2018) (SRainbow), and SimPLe and HRainbow scores as reported by Kaiser et al. (2019).

Similarly to Kaiser et al. (2019), sample efficiency is evaluated based on a mean score in the low data regime of 100k interactions with the real environment (400k frames). This is again motivated by the fairness of comparison between SimPLe and OTRainbow. On top of that, we compare the overall performance of all models depending on the amount of available data using median human normal-

ized performance. I.e. we normalize agent scores on each game such that 0% is the performance of the random agent and 100% corresponds to human score.

# 6 ANALYSIS

Section 6.1 in detail analyzes data efficiency. Then, section 6.2 focuses on the long term performance. Overall, OTRainbow and SimPLe prove to be the best models for 100k-interactions-only settings, without the clear winner between the two. Not surprisingly, SRainbow leads in regards to the long term performance as it does not sacrifice exploration to achieve the best possible scores within the first 100k steps. When comparing SimPLe to the variations of Rainbow DQN with respect to computational complexity, SimPLe is orders of magnitude more expensive. As shown in section 4, using SimPLe increases ratio $r$ 126 times, while the most computationally demanding variation of Rainbow - OTRainbow - increases $r$ 32 times. Thus, when taking into an account only the reinforcement learning part, SimPLe already requires almost 4 times more network updates. On top of that, however, SimPLe has to perform expensive training of the world model. As reported by Kaiser et al. (2019), a full version of SimPLe takes more than three weeks on 100k data points to complete the training. Using the same amount of data, OTRainbow is able to finish within the first 24 hours[2].

## 6.1 DATA EFFICIENCY

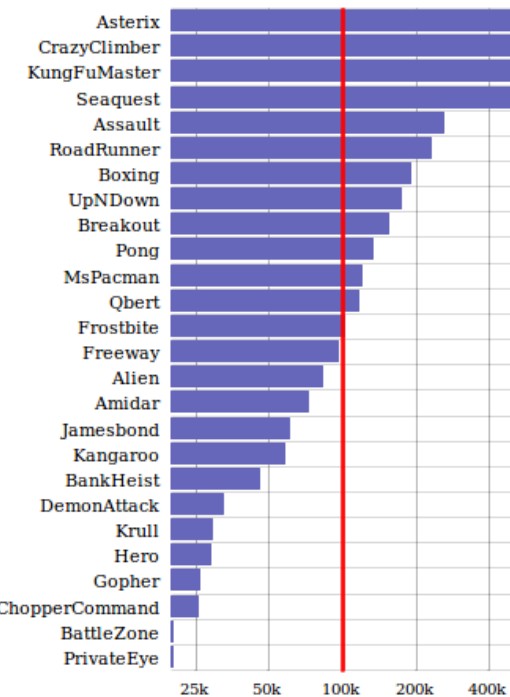

Figure 1: Comparison of SimPLe with OTRainbow. Bars represent the number of interactions required by OTRainbow to reach the same score as SimPLe achieves using exactly 100k interactions. Notice logarithmic scale on X-axis.

Results presented in this section are obtained after running 100k training interactions of the agent with the real environment (excluding Human). This setting is unfair towards SRainbow as it does not finish epsilon decay in that time. Nevertheless, its results are still provided as one of the baselines so it is clearly visible that although SRainbow is more likely to produce the best results in the long run, it achieves very poor performance during the first few iterations. Numerical results for this

---

[2]When running on 8 cores of Intel Haswell CPU.

setting are shown in Table 1. Moreover Figure 1 compares OTRainbow and SimPLe directly, using graphical convention similar to Kaiser et al. (2019). However, in this study, we use a logarithmic scale to denote the number of data samples needed to reach SimPLe's score. Doing so ensures that whether OTRainbow requires n times more experience or n times less, the visual absolute deviation from the SimPLe baseline is the same. Also, results are clipped to the absolute maximum deviation of 5x (i.e., 20k - 500k) as OTRainbow was evaluated on a maximum of 500k interactions due to computational constraints.

We can see that both OTRainbow and SimPLe outperform Random on all 26 games, interestingly neither HRainbow nor SRainbow managed to do the same. However, HRainbow falls behind Random only when playing Kangaroo. OTRainbow produces better scores than HRainbow on all games, it is a much better result than SimPLe's that manages to beat HRainbow only on 20 out of 26 games. In terms of direct comparison between OTRainbow and SimPLe, they perform very evenly. OTRainbow outperforms SimPLe on exactly half of the games but is dominated by SimPLe on the remaining half. Interestingly, the original paper behind SimPLe reported that efficiency on Freeway benefits most from the model-based approach, with SimPLe being 10x more efficient than HRainbow. However, this result is improved even further by OTRainbow as it manages to score over 8 points higher. We also calculate the median human normalized performance for each algorithm. Full numerical results of these calculations can be seen in Table 3 in Appendix A. Median human performance of OTRainbow beats SimPLe by over 10pp, however, SimPLe achieves super-human performance on 3 games (Pong, CrazyClimber, Boxing), while OTRainbow manages to do that only on Krull.

Overall, although both OTRainbow and SimPLe can learn much more optimal policies than all other models in the low-data regime, none of them visibly outperforms the other. These results show, that even the state-of-the-art model-based approach, highly tuned for achieving best scores given a small number of interactions with the real environment, is not significantly more data-efficient than slightly modified existing techniques.

## 6.2 DIFFERENT NUMBERS OF ITERATIONS

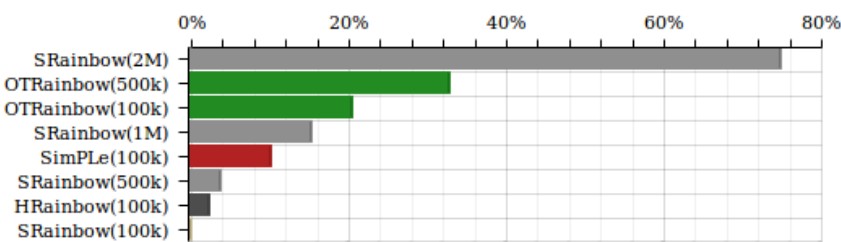

Figure 2: Median human normalized performance across all 26 games. Labels on the Y-axis specify the type of the algorithm used. Values in the brackets inform about the number of interactions before the method was evaluated. Bars represent the median result achieved by each of the approaches accordingly and are colorized depending on the type of algorithm used.

In addition to the score in the low data-regime, it is important that the agent can continue improving when performing any future interactions with the environment. To evaluate that, we tested OTRainbow in a settings with up to 500k interactions and provided SRainbow baseline for 500k, 1M, and 2M interactions. We were not able to execute experiments with a different number of real experience for SimPLe or HRainbow, the reasons being computational requirements of the former and undisclosed hyperparameters for the latter. However, we try to draw a comparison with SimPLe based on the analysis provided in the original paper.

Figure 2 shows the median human normalized performance for each of the evaluated methods. OTRainbow in both data settings scores surprisingly high, with its low-data regime version (100k) achieving better median result than SRainbow after full 1M steps. We hypothesize, however, that improvement of the performance of OTRainbow quickly slows down after the initial 500k steps, similarly to what was observed in SimPLe by Kaiser et al. (2019). This hypothesis is based on the change in performance between the 2 evaluations of OTRainbow, relatively to the standard algorithm. I.e., improvement between OTRainbow (100k) and OTRainbow (500k) is barely over 1.6x,

despite 5x more data. SRainbow, between the same data regimes, improves over 100x, which is followed by 5x improvement given only 2x more data twice (from 500k to 1M, and from 1M to 2M). Nevertheless, it should be confirmed empirically in future work.

## 7 CONCLUSION

We presented an intuition why the previous research did not use fair baselines when comparing new advancements with currently existing methods. We suggested the way of using state-of-the-art Rainbow DQN, namely OTRainbow, that leverages Rainbow's actual capabilities in terms of data efficiency. We experimentally proved that model-free OTRainbow is no worse than the state-of-the-art model-based approaches when given limited data while requiring an order of magnitude fewer computations. It shows that the recent work in sample efficient deep reinforcement learning does not produce significant improvements over the existing methods upholding the position of model-free algorithms as the state of the art, both in terms of data efficiency and total performance. Through these results, we aim to underline the importance of using appropriate model-free baselines, such as OTRainbow, in the future research that tries to improve data efficiency of deep RL approaches.

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

## A    COMPLETE NUMERICAL RESULTS

Table 2: Mean raw scores for each approach. Value in brackets after the name of the method indicates the number of training interactions performed before the evaluation.

| | OTRainbow (100k) | OTRainbow (500k) | SimPLe (100k) | HRainbow (100k) | SRainbow (100k) |
|---|---|---|---|---|---|
| Alien | 824.7 | 834.9 | 616.9 | 290.6 | 318.7 |
| Amidar | 82.8 | 215.3 | 74.3 | 20.8 | 32.5 |
| Assault | 351.9 | 549.3 | 527.2 | 285.7 | 231 |
| Asterix | 628.5 | 930.9 | 1128.3 | 300.3 | 243.6 |
| BankHeist | 182.1 | 223.9 | 34.2 | 34.5 | 15.5 |
| BattleZone | 4060.6 | 11093.8 | 4031.2 | 3363.5 | 3285.7 |
| Boxing | 2.5 | 8.4 | 7.8 | 0.9 | -24.8 |
| Breakout | 9.84 | 29.8 | 16.4 | 3.3 | 1.2 |
| ChopperCommand | 1033.33 | 1344 | 979.4 | 776.6 | 120 |
| CrazyClimber | 21327.8 | 28863.5 | 62583.6 | 12558.3 | 2254.5 |
| DemonAttack | 711.8 | 1303 | 208.1 | 431.6 | 163.6 |
| Freeway | 25 | 25.2 | 16.7 | 0.1 | 0 |
| Frostbite | 231.6 | 255.6 | 236.9 | 140.1 | 60.2 |
| Gopher | 778 | 748.5 | 596.8 | 748.3 | 431.2 |
| Hero | 6458.8 | 12461.3 | 2656.6 | 2676.3 | 487 |
| Jamesbond | 112.3 | 202.9 | 100.5 | 61.7 | 47.4 |
| Kangaroo | 605.4 | 3398 | 51.2 | 38.7 | 0 |
| Krull | 3277.9 | 3718.1 | 2204.8 | 2978.8 | 1468 |
| KungFuMaster | 5722.2 | 7261.7 | 14862.5 | 1019.4 | 0 |
| MsPacman | 941.9 | 1803.1 | 1480 | 364.3 | 67 |
| Pong | 1.3 | 19.9 | 12.8 | -19.5 | -20.6 |
| PrivateEye | 100 | 100 | 35 | 42.1 | 0 |
| Qbert | 509.3 | 8346.2 | 1288.8 | 235.6 | 123.4 |
| RoadRunner | 2696.7 | 6887.5 | 5640.6 | 524.1 | 1588.4 |
| Seaquest | 286.92 | 323.9 | 683.3 | 206.3 | 131.6 |
| UpNDown | 2847.6 | 4067 | 3350.3 | 1346.3 | 504.6 |
| | SRainbow (500k) | SRainbow (1M) | SRainbow (2M) | Human | Random |
| Alien | 481.5 | 766.3 | 1134.3 | 6875 | 184.8 |
| Amidar | 70.6 | 132.6 | 249.2 | 1676 | 12 |
| Assault | 468.6 | 630.1 | 1230.4 | 1496 | 249 |
| Asterix | 352.6 | 1038.7 | 2320.1 | 8503 | 234 |
| BankHeist | 17.5 | 304 | 872.1 | 734.4 | 15 |
| BattleZone | 3346.3 | 3453.7 | 11894.8 | 37800 | 2895 |
| Boxing | -29.5 | 8.3 | 47.1 | 4.3 | 0 |
| Breakout | 4.5 | 15.6 | 32.4 | 31.8 | 1 |
| ChopperCommand | 433.5 | 915.6 | 1810.1 | 9882 | 671 |
| CrazyClimber | 26090.9 | 66577.2 | 98461.7 | 35411 | 7339 |
| DemonAttack | 213.6 | 487.8 | 1748 | 3401 | 140 |
| Freeway | 8.2 | 27.45 | 31.9 | 29.6 | 0 |
| Frostbite | 275.2 | 512.3 | 2408.9 | 4335 | 74 |
| Gopher | 426.6 | 2119.2 | 3649.9 | 2321 | 246 |
| Hero | 326.6 | 3216 | 7875 | 25763 | 225 |
| Jamesbond | 50.2 | 236.1 | 472.2 | 406.7 | 29 |
| Kangaroo | 153.7 | 567.4 | 4252.9 | 3035 | 42 |
| Krull | 4714.2 | 6187.9 | 6136 | 2395 | 1543 |
| KungFuMaster | 596.7 | 10544.3 | 16284.5 | 22736 | 617 |
| MsPacman | 1244.2 | 1918.6 | 2301.5 | 15693 | 235 |
| Pong | -20.6 | -16.5 | 10.6 | 9.3 | -20 |
| PrivateEye | 692.8 | 169.1 | 92.5 | 69571 | 27 |
| Qbert | 450.6 | 1189 | 4046.9 | 13455 | 166 |
| RoadRunner | 1261.9 | 13793.9 | 31159 | 7845 | 0 |
| Seaquest | 181.2 | 378.4 | 1496.5 | 20182 | 61 |
| UpNDown | 1284.6 | 5566.3 | 10298.7 | 9082 | 488.4 |

Table 3: Mean human normalized score for each approach. Value in brackets after the name of the method indicates the number of training interactions performed before the evaluation.

| | OTRainbow (100k) | OTRainbow (500k) | SimPLe (100k) | HRainbow (100k) |
|---|---|---|---|---|
| Alien | 9.56% | 9.72% | 6.46% | 1.58% |
| Amidar | 4.27% | 12.23% | 3.76% | 0.54% |
| Assault | 8.27% | 24.09% | 22.32% | 2.96% |
| Asterix | 4.77% | 8.43% | 10.82% | 0.81% |
| BankHeist | 23.23% | 29.04% | 2.67% | 2.71% |
| BattleZone | 3.34% | 23.49% | 3.26% | 1.34% |
| Boxing | 55.00% | 202.50% | 187.50% | 15.00% |
| Breakout | 28.93% | 93.53% | 50.16% | 7.77% |
| ChopperCommand | 3.93% | 7.31% | 3.35% | 1.15% |
| CrazyClimber | 49.83% | 76.68% | 196.80% | 18.59% |
| DemonAttack | 17.53% | 35.66% | 2.09% | 8.94% |
| Freeway | 84.46% | 85.14% | 56.42% | 0.34% |
| Frostbite | 3.70% | 4.26% | 3.82% | 1.55% |
| Gopher | 25.64% | 24.22% | 16.91% | 24.21% |
| Hero | 24.41% | 47.91% | 9.52% | 9.60% |
| Jamesbond | 22.01% | 46.01% | 18.89% | 8.61% |
| Kangaroo | 18.82% | 112.13% | 0.31% | -0.11% |
| Krull | 203.66% | 255.35% | 77.67% | 168.55% |
| KungFuMaster | 23.08% | 30.04% | 64.40% | 1.82% |
| MsPacman | 4.57% | 10.14% | 8.05% | 0.84% |
| Pong | 73.06% | 135.69% | 111.78% | 3.03% |
| PrivateEye | 0.11% | 0.11% | 0.01% | 0.02% |
| Qbert | 2.58% | 61.56% | 8.45% | 0.52% |
| RoadRunner | 34.37% | 87.79% | 71.90% | 6.68% |
| Seaquest | 1.12% | 1.31% | 3.09% | 0.72% |
| UpNDown | 27.45% | 41.64% | 33.30% | 9.98% |
| **Median** | **20.42%** | **32.85%** | **10.17%** | **2.27%** |
| | SRainbow (100k) | SRainbow (500k) | SRainbow (1M) | SRainbow (2M) |
| Alien | 2.00% | 4.43% | 8.69% | 14.19% |
| Amidar | 1.24% | 3.53% | 7.26% | 14.27% |
| Assault | -1.43% | 17.62% | 30.57% | 78.70% |
| Asterix | 0.12% | 1.44% | 9.73% | 25.23% |
| BankHeist | 0.07% | 0.35% | 40.17% | 119.14% |
| BattleZone | 1.12% | 1.29% | 1.60% | 25.78% |
| Boxing | -627.50% | -745.00% | 200.00% | 1170.00% |
| Breakout | 0.97% | 11.65% | 47.57% | 101.94% |
| ChopperCommand | -5.98% | -2.58% | 2.66% | 12.37% |
| CrazyClimber | -18.11% | 66.80% | 211.02% | 324.60% |
| DemonAttack | 0.72% | 2.26% | 10.67% | 49.31% |
| Freeway | 0.00% | 27.70% | 92.74% | 107.77% |
| Frostbite | -0.32% | 4.72% | 10.29% | 54.80% |
| Gopher | 8.93% | 8.71% | 90.28% | 164.04% |
| Hero | 1.03% | 0.40% | 11.71% | 29.96% |
| Jamesbond | 4.82% | 5.56% | 54.81% | 117.35% |
| Kangaroo | -1.40% | 3.73% | 17.55% | 140.69% |
| Krull | -8.84% | 372.30% | 545.33% | 539.24% |
| KungFuMaster | -2.79% | -0.09% | 44.88% | 70.83% |
| MsPacman | -1.09% | 6.53% | 10.89% | 13.37% |
| Pong | -0.67% | -0.67% | 13.13% | 104.38% |
| PrivateEye | -0.04% | 0.96% | 0.20% | 0.09% |
| Qbert | -0.32% | 2.14% | 7.70% | 29.20% |
| RoadRunner | 20.25% | 16.09% | 175.83% | 397.18% |
| Seaquest | 0.35% | 0.60% | 1.58% | 7.13% |
| UpNDown | 0.19% | 9.27% | 59.09% | 114.16% |
| **Median** | **0.03%** | **3.63%** | **15.34%** | **74.77%** |

