# OpenReview forum: "Do recent advancements in model-based deep reinforcement learning really improve data efficiency?"
_ICLR.cc/2020/Conference — Reject_

### Official Review · AnonReviewer2 · 2019-10-22
**Official Blind Review #2**

**Rating:** 3

**Review:**

Do recent advancements in model-based deep reinforcement learning really improve data efficiency?

In this paper, the authors revisit the model-free baselines used in recent model-based reinforcement learning algorithms. And after more careful tuning of the hyperparameters, the model-free baselines can obtain comparable and much better performance using the same number of samples.

The paper is well-written, and the experiments are well-designed to support the claim.
However, the research contribution of the project is limited to image-space discrete RL tasks, and does not cover the wide-range other RL. In terms of the novelty, the proposed algorithm is not fundamentally different from Rainbow.
Therefore I tend to vote for borderline for this paper and am willing to increase the scores if more improvement is updated.

Besides, I would like to thank Mr. Ankesh Anand for mentioning the Hasselt et al. (2019) [1] paper, which is indeed very similar in terms of the topic discussed and the methods used to evaluate the algorithms.
I have also read Hasselt et al. (2019) before this submission, but I think it would be fair to say the two papers are relatively concurrent.

Potential improvement:
- It will be great if the authors also extend the discussion to current reinforcement learning algorithms that are applied in continuous tasks from states. In [2], similar conclusion is observed in continuous control tasks, where SAC [3] / TD3 [4] perform substantially better than many of the state-of-the-art model-based baselines.

[1] van Hasselt, Hado, Matteo Hessel, and John Aslanides. "When to use parametric models in reinforcement learning?." arXiv preprint arXiv:1906.05243 (2019).
[2] Wang, Tingwu & Bao, Xuchan & Clavera, Ignasi & Hoang, Jerrick & Wen, Yeming & Langlois, Eric & Zhang, Shunshi & Zhang, Guodong & Abbeel, Pieter & Ba, Jimmy. (2019). Benchmarking Model-Based Reinforcement Learning.
[3] Haarnoja, Tuomas, Aurick Zhou, Pieter Abbeel, and Sergey Levine. "Soft actor-critic: Off-policy maximum entropy deep reinforcement learning with a stochastic actor." arXiv preprint arXiv:1801.01290 (2018).
[4] Fujimoto, Scott, Herke van Hoof, and David Meger. "Addressing function approximation error in actor-critic methods." arXiv preprint arXiv:1802.09477 (2018).


**Experience Assessment:**

I have published in this field for several years.

**Review Assessment: Checking Correctness Of Derivations And Theory:**

I carefully checked the derivations and theory.

**Review Assessment: Checking Correctness Of Experiments:**

I carefully checked the experiments.

**Review Assessment: Thoroughness In Paper Reading:**

I read the paper thoroughly.

---

> ### Author Response · Authors · 2019-11-15
> **Response to Official Blind Review #2**
>
> Thank you for your thorough review and useful feedback. We identified two points raised in your comment.
>
>
> Point 1: The paper is limited to image-space discrete RL tasks, and does not cover the wide range of other RL.
>
> We partially addressed the second issue in the first point of our response to all reviewers. Please find the details there.
>
> In short, the problem was in the framing of the paper. It aimed to underline the importance of using fair baselines when proposing improvements in data efficiency of reinforcement learning algorithms and not necessarily compare MFRL to MBRL. We mainly found this issue in the studies that focus on image-space discrete RL.
>
>
> Point 2: The proposed algorithm is not fundamentally different from Rainbow.
>
> Indeed, that is the case. We did not intend to introduce a novel algorithm. Instead, our goal was to underline the importance of using appropriate baselines for data-efficient reinforcement learning by showing that already existing DQN-like algorithms can be much more data-efficient than it is often portrayed.

---

### Official Review · AnonReviewer3 · 2019-10-24
**Official Blind Review #3**

**Rating:** 3

**Review:**

The paper presents a data-efficient version of the Rainbow DQN by Hessel et al. (2018) that matches the performance of a recent state of the art model-based method by Kaiser et al. (2019) on several Atari games. Particularly, the paper empirically shows that a simple hyper-parameter tuning, in this case increasing the ratio of number of training steps to the environment interactions as well as decreasing epsilon-decay period, can result in significant improvements in sample efficiency of the Rainbow DQN agent. They show that their method (which requires significantly less computation) can outperform the model-based variant on half of the games tested, while performing worse on the rest.

Overall, I believe that this paper is below the acceptance threshold due to lack of 1) novelty, 2) significance and 3) depth of analysis.

The observation that more training updates with the agent’s existing experience results in  sample efficiency in DQN method has already been shown empirically by Holland et al. (2018) which has also been cited by this paper. Additionally, more recent work by Hasselt et al. (2019) which is not currently cited, explicitly addresses the same problem as this work by tuning the hyper-parameters of the Rainbow DQN to achieve significant sample efficiency, outperforming Kaiser et al. (2019) in 17 out of 26 Atari games tested. In addition, their work gave a rather detailed motivation and analysis of their findings, proposing and testing several hypotheses for how and when model-based methods could outperform replay-based model-free variants.

In comparison to prior work, the current paper has a more limited scope and significance. Hence, I believe more work would be needed to warrant acceptance.

Hessel et al., 2018: Rainbow: Combining Improvements in Deep Reinforcement Learning https://arxiv.org/abs/1710.02298
Kaiser et al., 2019: Model-Based Reinforcement Learning for Atari https://arxiv.org/abs/1903.00374
Holland et al., 2018: The Effect of Planning Shape on Dyna-style Planning in High-dimensional State Spaces https://arxiv.org/abs/1806.01825
Hasselt et al., 2019: When to use parametric models in reinforcement learning? https://arxiv.org/abs/1906.05243



**Experience Assessment:**

I have read many papers in this area.

**Review Assessment: Checking Correctness Of Derivations And Theory:**

N/A

**Review Assessment: Checking Correctness Of Experiments:**

I carefully checked the experiments.

**Review Assessment: Thoroughness In Paper Reading:**

I read the paper thoroughly.

---

### Official Review · AnonReviewer1 · 2019-10-24
**Official Blind Review #1**

**Rating:** 3

**Review:**

This paper presents an emprical study of how a properly tuned implementation of a model-free RL method can achieve data-efficiency similar to a state-of-the-art model-based method for the Atari domain.

The paper defines r as ratio of network updates to environment interactions to describe model-free and model-based methods, and hypothesizes that model-based methods are more data-efficient because of a higher ratio r. To test this hypothesis, the authors take Rainbow DQN (model-free) and modify it to increase its ratio r to be closer to that SiMPLe (model-based). Using the modified verison of Rainbow (OTRainbow), the authors replicate an experimental comparison with SiMPLe (Kaiser et al, 2019), showing that Rainbow DQN can be a harder baseline to beat than previously reported (Figure 1). This paper raises an important point about empirical claims without properly tuned baselines, when comparing model-based to model-free methods, identifying the amount of computation as a hyperparameter to tune for fairer comparisons.

I recommend this paper to be accepted only if the following issues are addressed. The first is the presentation of the empirical results. In Figure 1, OTRainbow is compared against the reported results in (Kaiser et al, 2019), along with other baselines, when limiting the experience to 100k interactions. Then, in Figure 2, human normalized scores are reported for varying amounts of experience for the variants of Rainbow, and compared against SiMPLe with 100k interactions, with the claim that the authors couldn't run the method for longer experiences. Unless a comparison can be made with the same amounts of experience, I don't see how Figure 2 can be interpreted objectively. In any case, the results in Figure 1 and the appendix are useful for showing that the baselines used in prior works were not as strong as they could be.

The second has to do with the interpretation of the results. The paper chooses a single method class of model-based methods to do this comparison, namely dyna-style algorithms that use the model to generate new data. But models can also be used for value function estimation (Model Based Value Expansion) and reducing gradient variance(using pathwise derivatives). The paper is written as if the conclusions could be extended to model-based methods in general. Can we get the same conclusions on a different domain where other model-based methods have been successful; e.g. continuous control tasks? A way to improve the paper would be to make it clear from the beginning that these results are about Dyna-style algorithms in the Atari domain.


**Experience Assessment:**

I have read many papers in this area.

**Review Assessment: Checking Correctness Of Derivations And Theory:**

N/A

**Review Assessment: Checking Correctness Of Experiments:**

I carefully checked the experiments.

**Review Assessment: Thoroughness In Paper Reading:**

I read the paper thoroughly.

---

> ### Author Response · Authors · 2019-11-15
> **Response to Official Blind Review #1**
>
> Thank you for your thorough review and useful feedback. Please find our response to the two issues raised below:
>
>
> Issue 1:
> In addition to comparing OTRainbow(100k) with SimPLe(100k), Figure 2 serves the analysis of the long term effects for each of the versions of Rainbow we introduced, similarly to what (Kaiser et al.) did in section 7.3. We do not aim to compare, e.g., OTRainbow(500k) with SimPLe(100k) because clearly, as you pointed out, it is an unfair comparison.
>
> However, given that we are currently extending our study to cover another recent advancement in the data-efficient reinforcement learning, analysis backed by this figure becomes less relevant. Therefore, as per your suggestion, we will delete it from the paper.
>
>
> Issue 2:
> We addressed the second issue in the first point of our response to all reviewers. Please find the details there.
>
> In short, as you have pointed out, the problem was in the framing of the paper. It aimed to focus on advancement in data efficiency for Atari (including MFRL for Atari), and not on MBRL as a whole.

---

### Public Comment · ~Ankesh_Anand1 · 2019-10-03
**Findings very similar to Hasselt et. al (2019)**

Thanks for the thorough investigation of sample-efficiency of model-free methods in the low-data regime. It seems Hasselt et. al (2019) [1] also reach the same conclusion: a well tuned Rainbow outperforms SimPLE in the low-data regime. They go on to add scenarios where MBRL could be useful.

It's nice that multiple groups reached the same conclusion as it adds credibility to the baseline. I am not sure about ICLR rules regarding this, but there could be an argument to consider this as concurrent work.

[1] van Hasselt, Hado, Matteo Hessel, and John Aslanides. "When to use parametric models in reinforcement learning?." To appear at NeurIPS '19. https://arxiv.org/abs/1906.05243

---

> ### Author Response · Authors · 2019-10-10
> **Thank you for sharing the preprint!**
>
> Thank you very much for your comment. Hasselt et. al (2019) indeed reaches a similar conclusion, thank you for pointing it out. We were not aware of this preprint before and the work was done fully concurrently.
>
> Although both studies achieve similar experimental outcomes, there are some important distinctions:
>
> 1. The high-level contrast between Hasselt et al. (2019) and our paper is that the former discusses differences between replay and model-based methods and the latter, first and foremost, emphasizes the importance of appropriate baselines in any future work focused on sample efficient deep reinforcement learning (not only model-based). We can agree that the title and related work section can be slightly misleading in that regard. It was motivated by the fact that the model-based approaches were producing the best results at the time of writing. One example of non-model-based work where arguments from this paper would be applicable could be Su Young et al. (2019) that proposed a novel model-free algorithm. It is not included as related work because we were not aware of any existing preprint at the moment of submission and did not want to base the analysis solely on abstract. We plan to include these points once the revision period opens.
>
> 2. Also, approaches to data-efficient hyperparameters of Rainbow DQN differ. Hasselt et. al (2019) increases the length of the multi-step update, whereas our paper increases the number of training steps per each data sample. Given that both studies employ simple and intuitive but different hyperparameter changes and do not exhaustively tune the algorithm, we believe that together they provide much stronger case behind using appropriate baselines showing that it does not take much to drastically improve the efficiency of existing methods.
>
> Lee, Su Young, Sungik Choi, and Sae-Young Chung. "Sample-efficient deep reinforcement learning via episodic backward update." To appear at NeurIPS '19 https://arxiv.org/abs/1805.12375
>
> van Hasselt, Hado, Matteo Hessel, and John Aslanides. "When to use parametric models in reinforcement learning?." To appear at NeurIPS '19. https://arxiv.org/abs/1906.05243

---

### Author Response · Authors · 2019-11-15
**Response to the most common points raised by the reviewers**

We want to thank all reviewers for all the time spent on analyzing our papers and for the constructive feedback. It is very much appreciated. We'll try to summarize and address all the concerns below.


1. The paper is framed as an analysis of MBRL methods but only compares to the dyna-style algorithms.

We aimed to show that recent advancements in sample efficiency of DRL (as benchmarked by the Atari domain) are solely due to using invalid baselines. I.e., they tend to use Rainbow/DQN in it's original or hypertuned form from Hessel et al. (2018) or Mnih et al. (2015). This form, however, did not focus on data efficiency but can be very easily modified to do so (as presented by OTRainbow). We chose the model-based dyna-like SimPLe as an example because it showed the most impressive results in that area. Nevertheless, we did not want to focus on MBRL but on sample-efficient RL as a whole. We agree that both the title and parts of the body are misleading. Simply when writing this paper, the MBRL algorithms were the main studies we were aware of that reported meaningful improvements over the Rainbow DQN. However, the same point would apply to MFRL for Atari. As an example, Lee et al. (2019) proposed a novel way of using experience replay for DQN that improves DQN's data efficiency. However, while doing so, it also increases the ratio r, without doing the same for the baseline to ensure a fair comparison.

We're in the process of experimentally comparing EBU and Overtrained version of DQN, and we will revise our paper to make our aims clearer.


2. Other papers (Holland et al., 2018; van Hasselt et al., 2019) have already introduced similar ideas.

While we agree it is often the case, there are some missing points that our work tries to cover:

a) Holland et al. (2018) have already shown that more training updates with the agent's existing experience results in higher sample efficiency of DQN. Nevertheless, Kaiser et al. (2019) mention that SimPLe outperforms the DynaDQN (and thus it outperforms replay-based methods). We clearly show that it is not the case.

b) van Hasselt et al. (2019) is a concurrent study that focuses on a comparison of model-based and replay-based algorithms. While the conclusion is similar, our goal, first and foremost, is to underline the importance of using appropriate baselines when introducing more data-efficient algorithms, not to analyze differences between different types of methods. To adhere to our aim of establishing a fair evaluation of data efficiency, we will add the comparison between EBU and Overtrained DQN showing that the problem of wrong baselines is not only relevant to MBRL.

d) Given the points above, we believe our study adds an essential aspect to the discourse in data-efficient RL. Especially, given that novel algorithms, although significant in other areas, are praised for their substantial improvements of data-efficiency, even though well established, existing methods are not worse. EBU and SimPLe being an example.

---

### Author Response · Authors · 2019-11-15
**Paper has not been revised**

Unfortunately, we were not able to finish revising the paper in the given time and include additional evaluations. Thus, we'll proceed with the initial version of the manuscript.

---

### Decision · Program_Chairs · 2019-12-19

**Decision:**

Reject

**Comment:**

The paper makes broad claims, but the depth of the experiments is very limited to a narrow combination of algorithms.